# AniEE: A Dataset of Animal Experimental Literature for Event Extraction

**Dohee Kim**[*◇]    **Ra Yoo**[*♣]    **Soyoung Yang**[◇]    **Hee Yang**[♡]    **Jaegul Choo**[◇]

[◇]KAIST AI

[♣]Advanced Institute of Convergence Technology,

[♡]Kookmin University

{dohee1121, sy_yang, jchoo}@kaist.ac.kr

rayoo@snu.ac.kr, yhee6106@kookmin.ac.kr

## Abstract

Event extraction (EE), as a crucial information extraction (IE) task, aims to identify event triggers and their associated arguments from unstructured text, subsequently classifying them into pre-defined types and roles. In the biomedical domain, EE is widely used to extract complex structures representing biological events from the literature. Due to the complicated syntactic nature and specialized domain knowledge, it is challenging to construct biomedical EE datasets. Additionally, most existing biomedical EE datasets primarily focus on cell experiments or the overall experimental procedures. Therefore, we introduce AniEE, a NER and EE dataset concentrated on the animal experiment stage. We establish a novel animal experiment customized entity and event scheme in collaboration with domain experts. We then create an expert-annotated high-quality dataset containing discontinuous entities and nested events and evaluate the recent outstanding NER and EE models on our dataset. The dataset is publicly available at: https://github.com/domyown/AniEE under CC BY-NC-ND 4.0 license.

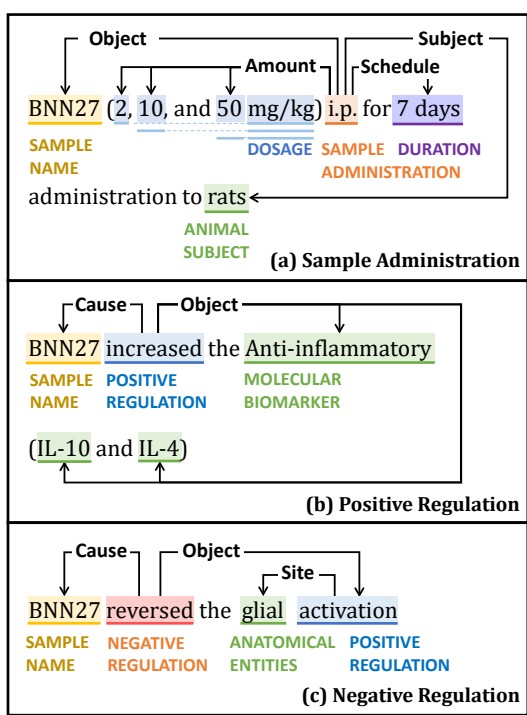

Figure 1: Each example for three event types from AniEE: SampleAdministration, PositiveRegulation, and NegativeRegulation

## 1 Introduction

Scientific literature can provide a groundwork for further investigations and offer a valuable platform for turning academic achievements into tangible industrial outcomes. Especially in the biomedical domain, the volume of literature grows exponentially (Frisoni et al., 2021)[1], posing challenges in locating relevant experimental key facts within unstructured text. To help researchers discover related literature that fulfills their needs, information extraction (IE) is actively utilized. Across diverse IE tasks, event extraction (EE) has been spotlighted to extract the complex syntactic structures of biological events. Each event consists of an event trigger and event arguments, as well as the relations between them. Figure 1 (a) illustrates an example of three SampleAdministration events, which includes an event trigger "i.p." and Object, Subject, Schedule, and three Amount arguments. Additionally, Figure 1 (b) and (c) describe examples of PositiveRegulation and NegativeRegulation, where "BNN27" is the Cause of the events.

Among various biomedical literature, animal experiment-related articles are one of the most difficult texts to extract valuable information from. The experimental phases can be generally divided into three stages: within cells, animal experiments, and clinical trials. Compared to cell experiments, animal research re-

---

[*]Equal Contributions.

[1]As of January 2021, PubMed alone has a total of 32M articles dating back to the year 1781. In 2020, three papers were published every minute.

| Dataset | Topic | Task | | Sub-task | | Number of Types | | | Count | | |
|---|---|---|---|---|---|---|---|---|---|---|---|
| | | NER | EE | Disc.Entity | Nest.Event | Entity | Role | Event | Entity | Role | Event |
| ShARe13 | Clinical notes | ✔ | | ✔ | | 1 | - | - | 11,161 | - | - |
| CADEC | Medical forum | ✔ | | ✔ | | 5 | - | - | 6,318 | - | - |
| GE11 | Cell - proteins | ✔ | ✔ | | ✔ | 2 | 6 | 9 | 16,976 | 10,270 | 14,840 |
| MLEE | Cell, Animal, Clinical trials | ✔ | ✔ | | ✔ | 16 | 9 | 26 | 8,291 | 7,588 | 5,554 |
| AniEE (Ours) | Animal | ✔ | ✔ | ✔ | ✔ | 12 | 5 | 3 | 22,105 | 17,538 | 10.546 |

Table 1: Comparison of two clinical-domain named entity recognition (NER) datasets and two biomedical-domain event extraction (EE) datasets: CLEF eHealth Task 2013 (ShARe13) (Danielle et al., 2013), CADEC (Sarvnaz et al., 2015), GENIA 2011 (GE11) (Kim et al., 2011), and Multi-level Event Extraction (MLEE) (Pyysalo et al., 2012). To the best of our knowledge, AniEE is the first dataset on animal experiment literature containing both discontinuous entities (Disc.Entity) and nested events (Nest.Event). The number of documents and sentences is described in Appendix Table 12.

quires further due diligence because they must consider ethical guidelines and have extensive resource requirements. More importantly, before moving on to further clinical trials with human participants, animal research serves as a significant step to evaluate safety and efficacy. Therefore, thorough investigations of previous research are essential to design animal experiments, verifying information such as species, dosage, and duration, and the relations between them, as shown in Figure 1.

Despite the importance of experimental information, EE studies targeting animal experiment literature have rarely been conducted. One of the reasons is, as described in Table 1[2], existing EE datasets contain literature that is either limited to the cell experiment stage (Kim et al., 2011; Pyysalo et al., 2013; Ohta et al., 2013) or does not specify a concrete experimental stage (Pyysalo et al., 2012). Therefore, an entity and event scheme that do not align with the dataset scope make it difficult to identify specific event triggers and their associated arguments, which are prevalent in animal experiments.

Therefore, we introduce AniEE, a named entity recognition and event extraction dataset focused on animal experiments. We establish a new entity and event scheme designed for the animal experiment stage in collaboration with domain experts. Our dataset is annotated by biomedical professionals and applied to the latest competitive NER and EE models. As described in Table 1, the novelty of our dataset lies in two aspects: addressing both 1) discon-

tinuous entities and 2) nested events. Existing benchmark datasets do not include both discontinuous entities and nested events. Therefore, we anticipate our dataset will contribute to the advance of all relevant NER and EE sub-tasks.

We sum up our contributions as follows:

- We introduce a new high-quality dataset AniEE consisting of 350 animal experimental literature annotated by domain experts.
- We define a novel entity and event scheme tailored to animal experiments.
- We evaluate the recent NER and EE approaches on our dataset dealing with both discontinuous entities and nested events.

## 2 Related Work

### 2.1 Complex Information Extraction

Traditional NER and EE formulate their tasks as a sequence labeling problem (Lample et al., 2016; Liu et al., 2018; Lin et al., 2019; Cao et al., 2019), assigning a tag to each token from a pre-defined tagging scheme (e.g., BIO tagging scheme). When faced with challenging syntactic scenarios such as nested or discontinuous text structures, the tagging scheme lacks the flexibility to address such complexities adequately. Consequently, alternative approaches have been explored for each task.

To handle sophisticated NER sub-tasks, 1) span-based methods (Luan et al., 2019; Wadden et al., 2019; Yu et al., 2020; Yamada et al., 2020), 2) hypergraph-based methods (Lu and Roth, 2015; Muis and Lu, 2016; Katiyar and Cardie, 2018; Wang and Lu, 2018), and 3) sequence generation-based methods (Straková et al., 2019; Yan et al., 2021; Fei et al., 2021)

---

[2]In Table 1, note that the number of events, relations, and entities are the sum of train and valid datasets in the case of GE11, CG, and PC because the test datasets of them are not available publicly. Also, if the number of trigger types is not equal to the number of event types in the dataset, we count the trigger type as the event type.

| Entity type | Frequency | Ratio (%) | Definition |
|---|---|---|---|
| SampleName | 4,566 | 20.7 | Include both inducers which promote a certain action, as well as inhibitors which suppress a certain activity |
| SampleType | 114 | 0.5 | Refer to the nature of the sample, including extract, oil, and powder |
| Dosage | 497 | 2.2 | Amount of sample administered to animals |
| Duration | 186 | 0.8 | Total period of sample administration to animals, excluding the animal handling period |
| DosageFrequency | 52 | 0.2 | Interval of sample administration |
| AnimalSubject | 1,199 | 5.4 | Animal species |
| AnimalStrain | 233 | 1.1 | Subtypes or genetic variants of animal species |
| AnimalSex | 102 | 0.5 | Sex of animal species |
| Anatomy | 3,514 | 15.9 | Body components, such as organs and tissues |
| MolecularBiomarker | 3,699 | 16.7 | Quantitative or qualitative measurement indicators of cellular-level biological process |
| Response | 6,323 | 28.6 | Physiological changes or responses associated with MolecularBiomarker |
| DiseaseName | 1,620 | 7.3 | Target disease investigated in a study |
| **Total** | 22,105 | 100.0 | |

Table 2: Definition and frequency of entity types. Ratios are presented rounded to the second decimal place.

have been proposed. Luan et al. (2019) enumerate all possible spans and recognize overlapped entities by span-level classification. Lu and Roth (2015) represent each sentence as a hypergraph with nodes indicating entity types and boundaries for overlapped NER. Muis and Lu (2016) extend this effort to apply discontinuous NER. Straková et al. (2019) use the sequence-to-sequence model to output a label sequence with multiple labels for a single token for overlapped NER.

Due to the syntactic complexity, research has focused on either nested or discontinuous entities. Recently, unified NER models (Li et al., 2020b, 2021, 2022) have been proposed to jointly extract these entity types.

On the other hand, to address advanced EE subtasks, pipeline-based methods (Li et al., 2020a; Sheng et al., 2021) have been introduced which sequentially extract event triggers and arguments. However, due to the inherent problem of error propagation in such sequential processes, OneEE (Cao et al., 2022) propose the one-step framework to simultaneously predict the relationship between triggers and arguments.

## 2.2 Clinical and Biomedical Datasets

Among the various domains where information extraction research is conducted, the clinical and biomedical domains are highly active fields based on numerous training datasets. In the clinical domain, ShARe13 (Danielle et al., 2013) and CADEC (Sarvnaz et al., 2015) are clinical report datasets including discontinuous drug event and disease entities. In the biomedical domain, several datasets (Pyysalo et al., 2009; Ohta et al., 2010) are derived from GE-NIA corpus (Ohta et al., 2002; Kim et al., 2003), including JNLPBA (Kim et al., 2004).

As a biomedical domain dataset for EE task, GE11 (Kim et al., 2011) is focused on events related to the transcription factors in human blood cells. CG (Pyysalo et al., 2013) is concentrated on the biological processes involved in the development of cancer. MLEE (Pyysalo et al., 2012) extend throughout all levels of biological organization from the molecular to the organism level, for both NER and EE tasks. In short, existing EE datasets consist of literature that is either restricted to cell experiments or generalized to cover all experimental stages. Hence, we introduce a novel dataset, AniEE, which aims to extract key animal experimental information.

## 3 Dataset Construction

### 3.1 Dataset Collection

The raw corpus was collected from PubMed, which is a widely-used online database containing a vast collection of biomedical literature. We collaborated with two senior domain experts to define the search terms, with the aim to crawl a diverse set of animal experimental literature. Medical subject headings (MesH), which serve as a hierarchical search index for the PubMed database, were used to determine the search term. First, we included ([MesHTerms] AnimalDiseaseModel OR [MesHTerms] Animals) in the search term to retrieve literature that performed animal experiments. Then, to obtain a literature collection on a wide range of topics, we set the search terms of ([MesHSubheading] Physiopathology OR [MesHSubheading] Chemistry) to include the literature which falls under physiopathology or chemistry as broad categories, while we excluded review articles. Thus, the final search terms consisted of ([MesHTerms] AnimalDiseaseModel OR

| Event Type | Argument Role | Definition | Freq. | Ratio (%) |
|---|---|---|---|---|
| SampleAdministration | Object, Subject, Site, Amount, Schedule | Administration of a specific sample to the experimental subject, including injection, oral administration, and topical application | 1,364 | 12.9 |
| PositiveRegulation | Object, Cause, Site | Stimulation of a biological process or system in animals that increases the activity, expression, or response of a particular target. | 5,811 | 55.1 |
| NegativeRegulation | Object, Cause, Site | Suppression or inhibition of a biological process or system in animals, resulting in reduced activity, expression, or response of a specific target. | 3,371 | 32.0 |

Table 3: Definition of event types and their corresponding argument roles. Ratios are presented rounded to the second decimal place.

[MesHTerms] Animals) AND ([MesHSubheading] Physiopathology OR [MesHSubheading] Chemistry) AND (Not Review). We collected the titles and abstracts of the literature from the top search results. We then removed the articles without the direct involvement of animal experiments, resulting in a total of 350 articles.

## 3.2 Entity and Event Type Scheme

AniEE contains 12 entity types and 3 event types. Table 2 and Table 3 describe the entity types and event types in our dataset, respectively. The event arguments are detailed in Appendix Table 13.

**Entity Types** Existing benchmark datasets (Ohta et al., 2002; Pyysalo et al., 2012) have typically focused on anatomical information, encompassing various entity types, especially at the cellular level. Given our primary focus on animal experiments, we consolidated these various types into a single unified category named Anatomy. On the other hand, the animal types are more fine-grained: AnimalSubject as a general identifier (e.g., "mice") and AnimalStrain as a specific identifier (e.g., "C57BL/6J"). Also, we introduce numerical entities, which are key attributes of the animal experiment design, such as Dosage, Duration, and DosageFrequency.

**Sample Administration** We annotate SampleAdministration on the text spans representing a specific approach to administering a sample. It can be a verb, such as a typical event trigger, but it can also be a non-noun part of speech, including a specialized abbreviation (e.g., "i.c.v") or an adverb (e.g., "orally"). When literature explicitly describes a method of administering a sample, we prioritize annotating specific expressions over generic verbs (e.g., "administered" and "treated"), as illustrated in Figure 1 (a). The event has five ar-

gument roles [3], including our two novel event argument roles to express the relations between the event trigger and arguments linked to experimental conditions (i.e., Dosage, Duration).

**Positive and Negative Regulation** We annotate PositiveRegulation and NegativeRegulation on the text spans that induce or interfere with biological processes. PositiveRegulation, such as the increase in cancer cells, does not necessarily mean an ultimately positive effect, and the same concept applies to NegativeRegulation. A unique characteristic of our dataset is that it contains *nested events*. Figure 1 (c) describes an example of nested events where the event PositiveRegulation ("activation") is the Object argument of another event NegativeRegulation ("reversed").

## 3.3 Annotation Procedure & Strategy

### 3.3.1 Annotation Procedure

To maintain annotation consistency between annotators, our annotation process consisted of two stages: pilot annotation and expert annotation. All annotators were instructed with detailed annotation guidelines [4].

**Pilot Annotation** Prior to performing the actual annotation process, a pilot annotation was conducted to train the annotators due to their varying levels of domain knowledge. It was performed to apply our newly defined scheme on 10 pieces of literature, which are extracted if they are related to the animal experiment stage from the MLEE corpus (Pyysalo et al., 2012), a publicly available biomedical event extraction dataset. During the annotation, we developed the annotation guidelines, and the annotators became familiar with the task

---

[3] Object refers to Theme in previous work.
[4] The annotation guidelines and examples are released at https://github.com/domyown/AniEE.

and tagtog [5], a web-based annotation tool with a user-friendly interface.

**Expert Annotation**  Six domain experts were recruited as annotators who are masters or Ph.D. candidates in the biomedical field. Two annotators independently annotated each piece of the literature.

### 3.3.2 Annotation for Complex Scenarios

Traditional annotation methods focus on continuous text spans, making it difficult to annotate certain entities and events due to the complex semantic nature of the animal experiment literature. To address this issue, we developed specialized annotation strategies to handle two specific scenarios: 1) the occurrences of discontinuous entities and 2) the instances where a solitary event trigger indicates several events.

**Discontinous Entity**  As shown in the Dosage case in Figure 1 (a), for numerical entity types, a substantial amount of literature list several numbers and then mention their unit only once at the end, resulting in discontinuous entities. To minimize the annotators' errors, these entities were subdivided into numeric (e.g., "8") and unit (e.g., "mg/kg") entities during the annotation process with a special relation type only for mapping the number and unit. We then post-process to merge them into a single entity (e.g., "8 mg/kg"). For Dosage, because the daily dosage units can be described, we temporarily segmented the unit entity into two unit sub-entities: DosageUnit (e.g., "mg/kg") and DosageDayUnit (e.g., "/day"), which were later combined into one (e.g., "8 mg/kg /day").

**Multiple Events on a Single Event Trigger**  Given an example of "ginsenoside Rb1 (35 mg/kg) and losartan (4.5 mg/kg) i.c.v", an event trigger ("i.c.v") has two samples and corresponding dosages for each sample. Since an event trigger corresponds to only a single instance of an Object argument, the example represents one event for each sample, with a total of two events. However, prior research has found that these scenarios are challenging to extract due to the inherent semantic intricacy, which has consequently been acknowledged as a limitation (Friedrich et al., 2020). In order to accurately extract these events, we introduce a supplementary relation type to link each dosage associated with each respective sample (e.g.,

| Statistics | Train | Valid | Test |
|---|---|---|---|
| Number | 250 | 50 | 50 |
| Avg.Sent | 11.6 | 11.6 | 10.8 |
| Avg.Token | 455.5 | 474 | 429.3 |
| Avg.Entities/Doc | 83.5 | 86.4 | 76.1 |
| Avg.Events/Doc | 30.4 | 31.3 | 28.5 |

Table 4: Statistics of AniEE. The average length of sentences per document (Avg.Sent), the average number of tokens per sentence (Avg.Token), the average number of entities (Avg.Entities/Doc), and events per document (Avg.Events/Doc) are reported.

| Tasks | P | R | F1 | IAA |
|---|---|---|---|---|
| NER | 0.943 | 0.947 | 0.944 | 0.973 |
| EE | 0.687 | 0.656 | 0.662 | 0.586 |

Table 5: Annotation quality related scores of 10 randomly sampled documents. Precision (P), recall (R), and F1 score are macro averaged, and inter-annotator agreement (IAA) is Krippendorff's $\alpha$.

between "losartan" and "4.5 mg/kg") and instruct the annotators. After the annotation process, post-processing is conducted to produce two distinct events for each sample.

### 3.4 Dataset Statistics and Characteristics

The AniEE corpus contains a total of 350 animal experimental articles. We split our dataset into 4:1:1 for the training, validation, and test sets. Table 4 [6] presents their statistics.

**Named Entity Types**  Table 2 shows the long-tail distribution of the frequency of the entity types in our dataset similar to other datasets (Pyysalo et al., 2012; Luo et al., 2022), including Response (28.6%), SampleName (20.7%), MolecularBiomarker (16.7%), and Anatomy (15.9%). Our dataset contains 644 discontinuous entities.

**Event Types**  Table 3 presents the frequency of event types in our dataset, such as SampleAdministration (12.9%), PositiveRegulation (55.1%), and NegativeRegulation (32.0%). In addition, for PositiveRegulation and NegativeRegulation event types, since an event can be a possible argument for another event, AniEE has 709 (6.7%) nested events.

---

[5] www.tagtog.com

[6] BioBERT from https://huggingface.co/dmis-lab/biobert-v1.1

**IAA** To evaluate the annotation quality, we report a standard measure of the inter-annotator agreement (IAA) of Krippendorff's $\alpha$ (Krippendorff, 2011), as well as precision, recall, and F1 measure. We estimate each score for two tasks: NER and EE tasks on ten randomly-sampled documents. Each document is independently annotated by two annotators: one Ph.D. candidate and one master's student. We consider the annotation labels of the Ph.D. candidate as the ground truth and compare them to the other one's. Table 5 shows the IAA results. Krippendorff's $\alpha$ are 0.973, and 0.586 for the NER and EE tasks, respectively. Morante et al. (2009) has reported that the IAA scores of Sasaki et al. (2008) and GENIA event corpus are 0.49, and 0.56. These low scores were attributed to the difficulty of the annotation due to the complexity of the EE dataset structure. Compared to these scores, our IAA scores are favorable.

## 4 Experiments

### 4.1 Settings

To examine the effectiveness and challenges of AniEE corpus, we conduct experiments on recent superior baseline models for the NER and EE tasks.

**NER Models** We evaluate our dataset on unified NER models, which allow us to extract discontinuous and flat entities. **W2NER** (Li et al., 2022) is an unified NER framework by modeling 2D grid matrix of word pairs. **SpanNER** (Li et al., 2021) proposes a span-based model which extracts entity fragments and the relations between fragments to jointly recognize both discontinuous and overlapped entities.

**EE Models** Given the nascent nature of the nested event extraction (EE) task, only a limited number of approaches have been introduced in recent years. In our study, we assess our dataset using end-to-end models, wherein both event triggers and arguments are predicted simultaneously. The current methodologies encompass two major categories: pipeline-based multi-staged approaches and one-staged approaches. **CasEE** (Sheng et al., 2021) sequentially type detection task, followed by trigger and argument detection task with a cascading decoding strategy. **OneEE** (Cao et al., 2022) simultaneously perform event extraction task using word-word relation matrix. All models are sentence-level baselines, but we extend the baseline input to the document level, feeding multiple sentences as input, to predict the events based on a

| Baselines | P | R | F1 |
|---|---|---|---|
| SpanNER | 66.84 | 71.88 | 69.27 |
| W2NER | 72.24 | 69.64 | 70.92 |

Table 6: NER performance on two baselines: W2NER (Li et al., 2022) and SpanNER (Li et al., 2021). All scores are the percentage (%).

| W2NER | P | R | F1 |
|---|---|---|---|
| *Sample* | | | |
| SampleName | 80.67 | 79.11 | 79.88 |
| SampleType | 66.67 | 50.00 | 57.14 |
| *Dose* | | | |
| Dosage | 77.27 | 77.27 | 77.27 |
| Duration | 57.14 | 70.59 | 63.16 |
| DosageFrequency | 60.00 | 25.00 | 35.29 |
| *Animal* | | | |
| AnimalSubject | 89.39 | 93.57 | 91.43 |
| AnimalStrain | 61.70 | 90.62 | 73.41 |
| AnimalSex | 1.0 | 69.23 | 81.82 |
| *Target* | | | |
| Anatomy | 71.52 | 74.27 | 72.87 |
| MolecularBiomarker | 77.06 | 69.56 | 73.12 |
| Response | 58.99 | 61.02 | 59.99 |
| DiseaseName | 76.35 | 78.06 | 77.20 |

Table 7: NER performance of W2NER (Li et al., 2022) for each entity type. The entity types are grouped by *Gray categories* for readability.

better understanding of contextual representations. Since several documents are longer than the maximum sequence length of BioBERT, we split the document into smaller chunks.

**Evaluation Metric** Following previous work in the NER and EE tasks (Sheng et al., 2021; Cao et al., 2022), we report Precision (P), Recall (R), and F measure. We measure four evaluation metrics for the EE task. 1) Trigger Identification: an event trigger is correctly identified if the predicted trigger matches with a ground truth; 2) Trigger Classification: an event trigger is correctly classified if the identified trigger is identical to the right event type; 3) Argument Identification: an event argument is correctly identified if the predicted argument aligns with a ground truth; 4) Argument Classification: an event argument is correctly identified if the identified argument is assigned to the right role. In short, we can organize identification as the task of finding the region of a trigger or entity and classification as predicting the class of each object.

| Models | Trigger Identification | | | Trigger Classification | | | Argument Identification | | | Argument Classification | | |
|---|---|---|---|---|---|---|---|---|---|---|---|---|
| | P | R | F1 | P | R | F1 | P | R | F1 | P | R | F1 |
| CasEE | 70.83 | 73.54 | 72.16 | 67.56 | 70.23 | 68.87 | 54.07 | 59.19 | 56.51 | 53.31 | 58.54 | 55.81 |
| OneEE | 68.98 | 34.82 | 46.28 | 67.25 | 33.95 | 45.12 | 59.04 | 29.42 | 39.47 | 60.23 | 26.40 | 36.71 |

Table 8: Event extraction performance comparison of two baseline models for the four evaluation metrics: trigger identification, trigger classification, argument identification, and argument classification (see Section 4.1).

**Implementation Detail** We use BioBERT (Lee et al., 2020) as a pretrained language model and adopt AdamW optimizer (Ilya and Frank, 2019) with a learning rate of 2e-5. The batch size is 8, and the hidden size $d_h$ is 768. We train all baselines with 200 epochs.

## 4.2 Experimental Results

### 4.2.1 Named Entity Recognition

Table 6 shows the precision, recall, and micro-average F1 scores of the NER baselines. W2NER slightly outperforms SpanNER based on F1 score.

**Prediction of Entity Types** Table 7 presents the precision, recall, and F1 scores of W2NER for each entity type. We assume that the long-tail frequency distribution and the number of unique expressions within each entity type affect the performance. Head-frequency entity types, such as SampleName and DiseaseName, where their frequencies are 4,566 and 1,620, show higher performance than tail entity types, such as SampleType and DosageFrequency, where their frequencies are 114 and 52, respectively. Also, the F1 scores of AnimalSubject and AnimalSex are higher than other entity types. This is because the concepts of animal subjects and sex are less specific to animal experiments than other entity types, making it easier to leverage the knowledge gained from the pretrained language model.

### 4.2.2 Event Extraction

Table 8 shows the EE results for the four evaluation metrics. CasEE consistently outperforms OneEE, except for the precision scores for the argument identification and classification metrics. Overall, CasEE has a higher recall than precision across all metrics, which suggests that the model produces more false positive predictions than false negatives. On the other hand, OneEE has a large gap between precision and recall across all metrics, which implies that the model generates a lot of false negatives and easily misses gold triggers and entities.

| CasEE | P | R | F1 |
|---|---|---|---|
| SampleAdministration | 56.56 | 64.49 | 60.26 |
| PositiveRegulation | 65.32 | 68.88 | 67.05 |
| NegativeRegulation | 76.89 | 74.81 | 75.83 |

Table 9: Event extraction performance of the triggers for each event type.

| CasEE | P | R | F1 |
|---|---|---|---|
| *SampleAdministration* | | | |
| Object | 54.68 | 69.09 | 61.04 |
| Subject | 61.54 | 80.00 | 69.57 |
| Site | 83.33 | 71.43 | 76.92 |
| Amount | 67.80 | 90.91 | 77.67 |
| Schedule | 83.33 | 40.00 | 54.05 |
| *PositiveRegulation* | | | |
| Object | 47.54 | 53.49 | 50.34 |
| Cause | 60.33 | 70.29 | 64.92 |
| Site | 34.29 | 30.77 | 32.43 |
| *NegativeRegulation* | | | |
| Object | 53.47 | 51.25 | 52.33 |
| Cause | 59.87 | 73.39 | 65.94 |
| Site | 15.38 | 11.11 | 12.90 |

Table 10: Event extraction performance of the arguments for each *event type*.

**Extraction of the Event Types** Table 9 shows the precision, recall, and F1 score for each event type. This is the result when the model proceeds to trigger identification and classification jointly. The low accuracy of SampleAdministration, where the frequency ratio is 12.9%, can be explained by the low frequency compared with other event types. However, although PositiveRegulation (55.1%) appears more frequently than NegativeRegulation (32.0%), the model predicts NegativeRegulation more accurately than PositiveRegulation. Hence, we further analyze this observation in Section 5.1.

**Extraction of the Event Arguments** Table 10 describes the extraction performance of the event arguments for each event type. This is the result when the model predicts the argu-

ment identification and classification together. CasEE (Sheng et al., 2021) shows the highest F1 score on the Amount in SampleAdministration. This result can be explained by the fact that the mentions of Amount usually consist of a numeric and a specific unit component, making it easier for the model to detect this consistent pattern in unstructured documents.

In addition, the F1 scores of Site in PositiveRegulation and NegativeRegulation are significantly lower compared to SampleAdministration. Site in SampleAdministration typically refers to the location of administration and has a low variance. In contrast, Site in regulation refers to where the effect occurs and therefore includes a wider range of locations than the location of administration. Therefore, Sites in PositiveRegulation and NegativeRegulation can be described as more difficult to detect compared to SampleAdministration because of the variety of places where effects occur.

**Recognition of the Event Trigger** On the EE task, the model needs to detect the span of the event trigger. We set this task to NER, which detects the span of the event trigger mention. As described in Table 11, W2NER (Li et al., 2022) shows progressively higher scores for NegativeRegulation, PositiveRegulation, and SampleAdministration. It is similar to the event trigger classification performance in Table 9; the order of score is the same, Thus, we can analyze that the low NER performance of PositiveRegulation is due to the imbalanced distribution of trigger mentions.

## 5 Analysis

### 5.1 Distribution of the Event Mentions

In the event trigger classification task, we expected that the model would predict PositiveRegulation more accurately than NegativeRegulation because the frequency of PositiveRegulation is 1.7 times greater than one of NegativeRegulation. However, the results shown in Table 9 contradict our expectations. To analyze this experiment, we collected the mentions of the event triggers for each event type and extracted the lemmas to group the mentions in a semantic manner [7]. To calculate the frequency ratio of a lemma cluster per total mention frequency, we summed the frequencies of all the mentions belonging to the cluster.

---

[7]We utilize NLTK WordNet Lemmatizer. Top-5 lemma cluster for each event type is described in Appendix A.2

| W2NER | P | R | F1 |
|---|---|---|---|
| SampleAdministration | 60.53 | 56.56 | 58.48 |
| PositiveRegulation | 71.06 | 60.28 | 65.23 |
| NegativeRegulation | 82.64 | 67.34 | 74.21 |

Table 11: NER performance scores when detecting the text span of the event triggers. All scores are the percentage (%).

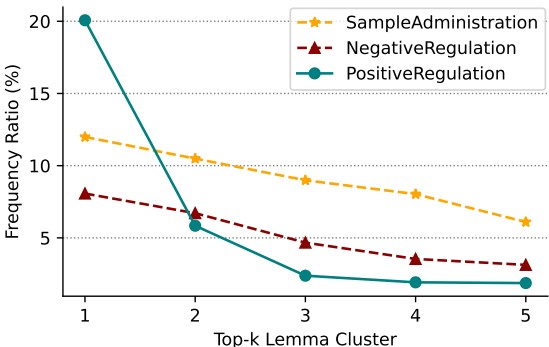

Figure 2: Freqeucny ratio comparison for each event type. Each line represents an event type. We plot the percentage of the event mention distribution (Y-axis) accounted for by the top five lemma clusters (X-axis).

As shown in Figure 2, in PositiveRegulation, the top-1 lemma cluster with the lemma "induced" accounts for 20% of the overall frequency ratio, while the second lemma cluster ("increased") accounts for 5.8%. This is distinguishable with other event types, such as SampleAdministration and NegativeRegulation, where the frequency percentage gradually decreases with each subsequent lemma cluster. Therefore, the low performance of PositiveRegulation in Table 9 can be explained by the imbalanced distribution of the trigger mentions.

## 6 Conclusion

In order to enhance the efficiency and accuracy of a comprehensive review of existing animal experiment literature, we introduce AniEE, an event extraction dataset designed specifically for the animal experimentation stage. The distinctiveness of our dataset can be summarized through two key points. To the best of our knowledge, our dataset represents the first event extraction dataset focusing on animal experimentation. In addition, our dataset encompasses both discontinuous named entities and nested events within texts at the document level. We anticipate that introducing our novel dataset, AniEE, will contribute significantly to advancing document-level event extraction not only in

the biomedical domain, but also in the natural language processing.

## Limitations

We acknowledge that our dataset, which contains 350 abstracts, is not huge due to labor-intensive manual annotation. However, considering the number of annotated events and entities, as well as our experimental results, the current dataset size is sufficient to develop NER and EE models.

## Ethics Statement

As each annotator is a master's or Ph.D. student, we compensated each annotator reasonably comparable to a lab stipend. Additionally, we adjusted the workload weekly, accounting for each annotator's schedule.

## Acknowledgement

This work was supported by a grant (21162MFDS076) from Ministry of Food and Drug Safety in 2023, Institute of Information & communications Technology Planning & Evaluation (IITP) grant funded by the Korea government (MSIT) (No.2019-0-00075, Artificial Intelligence Graduate School Program (KAIST)), and the National Research Foundation of Korea (NRF) grant funded by the Korea government (MSIT) (No. NRF-2022R1A2B5B02001913). Also, we would like to express our gratitude to Jiyun Lee, Heejeong Hwang, Jungyeon Yu, and Jungyeon Lee for their participation in this work.

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

# A Dataset Construction

## A.1 Dataset Statistic

| Dataset | Count | | | Corpus | |
|---|---|---|---|---|---|
| | Event | Role | Entity | Document | Sentence |
| ShARe13 | - | - | 11,161 | 298 | 18,767 |
| CADEC | - | - | 6,318 | 1250 | 7,597 |
| GE11 | 10,270 | 14,840 | 16,976 | 1,514 | 14,962 |
| MLEE | 5,554 | 7,588 | 8,291 | 262 | 2,607 |
| AniEE (Ours) | 9,140 | 14,151 | 21,973 | 398 | 4,581 |

Table 12: Dataset statistic comparison.

Table 12 shows the dataset comparison with existing benchmarks with corpus information. Also, we describe the definition and frequency of event arguments in Table 13.

## A.2 Top-5 Lemma Cluster of Event Type

As mentioned in Section 5.1, we collect the mentions of event triggers and group them by lemma based on NLTK WordNet Lemmatizer. For instance, if "activated by" is a trigger mention, the cluster of "activated" includes it. For each event type, the top-5 frequent lemma clusters with their total frequency and ratio (%) are as follows:

- **PositiveRegulation**: *induced* (752, 20.1%), *increased* (219, 5.8%), *increase* (89, 2.4%), *associated* (72, 1.9%), *improved* (70, 1.9%)
- **NegativeRegulation**: *reduced* (178, 8.1%), *decreased* (148, 6.7%), *anti* (103, 4.7%), *inhibitor* (78, 3.5%), *attenuated* (69, 3.1%)
- **SampleAdministration**: *administration* (112, 12.0%), *treated* (98, 10.5%), *injection* (84, 9.0%), *treatment* (75, 8.0%), *administered* (57, 6.1%)

# B Case Study

To further investigate 1) discontinuous entities and 2) nested events in our dataset, we visualize six samples of our dataset.

## B.1 Discontinuous Entity

We extract data samples that contain discontinuous entities, color the named entities with each color of their entity type, and tag whether the prediction of this entity is a success or fail. W2NER (Li et al., 2022) is utilized to extract model prediction. As shown in Table 14, the model predicts the discontinuous entities for the first three examples accurately. However, the model fails to detect the duration entity of the fourth example (i.e., "five days") since it predicts "five consecutive days" as a flat entity. This is because we define Duration as a number and unit in the annotation strategy.

## B.2 Nested Event

Similar to discontinuous entities, we color the event triggers in a given data sample and tag whether CasEE (Sheng et al., 2021) correctly predicts them. Also, we extract the relations between two event triggers when one of the triggers is an argument of the other. The relations between triggers are described by a triplet, where the first is the event trigger of the current example, the second is the argument of the first, and the third is the role of the second argument within the event of the first trigger. Table 15 shows two examples of nested events. The model shows incorrect prediction in the first example and correct prediction in the second example, but the argument roles are the same as for Object.

| Argument Role | Frequency | Ratio (%) | Definition |
|---|---|---|---|
| *SampleAdministration* | | | |
| Object | 1,366 | 41.90 | A material which is used for an event |
| Subject | 690 | 21.17 | An animal experimental subject of an event |
| Site | 240 | 7.36 | Body region where an event occurrs |
| Amount | 729 | 22.36 | Quantity measurement of a sample |
| Schedule | 215 | 6.59 | A time frame of an event |
| **Total** | **3,240** | **100.0** | |
| *PositiveRegulation* | | | |
| Object | 5,575 | 62.85 | Physiological parameters affected by an event "SampleAdministration" |
| Cause | 2,734 | 30.82 | Attribute that influences the modifications of the target factor |
| Site | 562 | 6.3 | Physiological region where an Object argument is observed |
| **Total** | **8,871** | **100.0** | |
| *NegativeRegulation* | | | |
| Object | 3,267 | 60.20 | Physiological parameters affected by an event "SampleAdministration" |
| Cause | 1,888 | 34.79 | Attribute that influences the modifications of the target factor |
| Site | 272 | 5.0 | Physiological region where an Object argument is observed |
| **Total** | **5,427** | **100.0** | |

Table 13: Definition and frequency of arguments for each *event type*. We sum up the frequency of arguments for each event type and get the ratio in the event type. Ratios are presented rounded to the second decimal place.

| Example | Discontinuous Entity | Prediction |
|---|---|---|
| Mice (Swiss Webster) were exposed to toluene (0, 2000 or 4000 ppm, 30 min a day) | 0 ppm
2000 ppm | **Success** |
| Female Sprague-Dawley rats were treated orally with an ascending methadone dosage schedule (5, 10, 15, 20, 25 and 30 mg/kg/day) | 5 mg/kg/day
10 mg/kg/day
15 mg/kg/day
20 mg/kg/day
25 mg/kg/day | **Success** |
| 3xTg mice were fed a control or Cr-supplemented (3% Cr (w/w)) diet for 8-9 weeks | 8 weeks | **Success** |
| The rats were treated with SeNPs by intraperitoneal injection (0.52009mg SeNP/kg) for five consecutive days | five days | **Fail** |
| **Types**: Dosage, Duration | | |

Table 14: Event examples of `AniEE` that contain discontinuous entities. Note that the second column only describes the discontinuous entities, excluding the flat ones, such as "30 min", "9 weeks", and "0.52009mg SeNP/k". The entity types are represented as colors for readability.

| Example | Nested Events | Prediction |
|---|---|---|
| the mechanical and metabolic disruption of cartilage was prevented in vivo. | (prevented, disruption, Object) | **Fail** |
| Protective and anti-inflammatory effect of selenium nano-particles against bleomycin-induced pulmonary injury in male rats | (against, protective, Object) | **Success** |
| **Types**: PositiveRegulation, NegativeRegulation | | |

Table 15: Nested event examples. The event types are represented as colors for readability. The relation between the event trigger and its argument is described in relation triplet: *(Trigger mention, Argument mention, Relation)*.