# OpenReview forum: "AniEE: A Dataset of Animal Experimental Literature for Event Extraction"
_EMNLP/2023/Conference — EMNLP 2023 Findings_

### Official Review · Reviewer_7ePo · 2023-07-27

**Soundness:** 4

**Excitement:**

4: Strong: This paper deepens the understanding of some phenomenon or lowers the barriers to an existing research direction.

**Paper Topic And Main Contributions:**

This paper introduces a new dataset for biomedical NER and EE. It focuses on scientific literature about animal experiments, a stage of clinical research for which the authors indicate there aren’t as many IE resources as cell stage research. The dataset includes 350 documents manually annotated with 22k entities and 10k events.

The authors developed their own schema of entity and event types relevant for animal experiments, consulting with domain experts. Notably, the dataset contains discontinuous entities and nested events, which the authors found to be important structures for capturing specific attributes of the relevant experiments.

The paper focuses mainly on the introduction of the dataset, a detailed explanation of the process of collecting and annotating the included examples, and resulting frequencies of events and entities. The paper also includes experimental results from training several existing NER and EE models on the new dataset, showing that the dataset can be used to train models to detect and classify the included entities and events.

**Reasons To Accept:**

The new dataset seems like a valuable contribution to an area of biomedical IE where there seems to be demand and a potential positive impact on other research. The authors provide useful examples of report excerpts about animal experiments, showing very technical language that might be difficult to parse and extract relevant details from, without labeled data of the same format.

At the same time, the components of these events are reasonably straightforward, with the authors defining just three event types — Sample Administration (a dosage of something was administered to some animal subject), Positive Regulation (the sample made something else increase) and Negative Regulation (the sample made something else decrease). All of those other “somethings” are extracted entities with defined roles in the event.

This suggests that information about animal experiments can be organized into consistent patterns, that allow for capturing many relationships between a wide variety of treatments and outcomes, in a way that could be easily quantified and analyzed. As far as event schemas go, this one seems to be appealingly well structured to cover many new types of experiments, and even a non-domain expert can imagine its useful applications.

The authors also describe a reasonably thorough process of developing their schema and annotating their documents. They demonstrate the flexibility of how they defined their discontinuous entities to extract dosage amounts, units, and per-time-period unit aspects to compose complete dosage information. They also describe a two-stage annotation process, to align their annotators on consistent guidelines. And they report sample inter-annotator agreement for both entities and events. The inter-annotator agreement for entities in particular is very high.

The experiments, while briefly described in this paper, also appear to show that the dataset can be used to train reasonably useful models for detecting and classifying the annotated entities and events in animal experiments.

**Reasons To Reject:**

There are a few points where the paper appears to be a bit weaker, although they may not be major weaknesses. One concern is that inter-annotator agreement on the event annotations (0.586) is much lower than for the entities (0.973). The authors cite IAA rates for another related dataset to show that theirs are higher, as justification that theirs are sufficient. But they don’t say anything further about why there was so much more disagreement about the event annotations.

Does this suggest an upper bound on the potential accuracy or F1 scores that a model trained on this dataset could achieve for event detection and/or classification? It is possible that events are much more difficult for humans to agree on. But the relative simplicity of their event types gives the impression that it might have been possible to do another round of aligning between the two annotators on the exact definitions of events to be annotated. A bit more discussion of why it wasn’t feasible to get higher agreement might help build confidence in the usefulness of that part of the dataset.

The model experiments at the end of the paper are also not described in much detail. In this case, that is probably fine, since the authors are reproducing existing models from past research, and the focus of their paper is on the new dataset rather than any model innovations. They show results for two models each for the NER and EE tasks, respectively. There isn’t a lot to conclude from those findings, though, other than that the models do seem to be learning.

I’m not entirely sure how to assess whether these results are impressive or not, because no comparative results from similar research are shown. The authors are the first to use their own dataset. But since their goal is to demonstrate the value of their new dataset, it actually might be helpful to show how the same types of models performed on a previous dataset. If the authors get at least similar scores when the models are trained on their own dataset, that could indicate that their dataset is as consistently labeled and useful for training NER and EE models as other datasets have been.

It would also be reasonable, however, to expect that different entity and event schemas will lead to different average performance of models trained on them. So model results from past research using other datasets might not be very comparable. Again, I don’t think this is a major issue, because the paper’s main focus is not the model experiments. But here too, a little more discussion of what we should learn from the experiments, or how they might suggest a jumping-off point for other researchers seeking to use this dataset, might help strengthen the paper more.

**Reproducibility:**

4: Could mostly reproduce the results, but there may be some variation because of sample variance or minor variations in their interpretation of the protocol or method.

**Reviewer Confidence:**

3: Pretty sure, but there's a chance I missed something. Although I have a good feel for this area in general, I did not carefully check the paper's details, e.g., the math, experimental design, or novelty.

---

> ### Author Rebuttal · Authors · 2023-08-29
>
> We sincerely appreciate the thoroughness invested in your review. We extend our gratitude for all constructive comments. To enhance readability, we have noted the comments in block quotes along with the corresponding responses below:
>
> > There are a few points where the paper appears to be a bit weaker, although they may not be major weaknesses. One concern is that inter-annotator agreement on the event annotations (0.586) is much lower than for the entities (0.973). The authors cite IAA rates for another related dataset to show that theirs are higher, as justification that theirs are sufficient. But they don’t say anything further about why there was so much more disagreement about the event annotations. Does this suggest an upper bound on the potential accuracy or F1 scores that a model trained on this dataset could achieve for event detection and/or classification? It is possible that events are much more difficult for humans to agree on. But the relative simplicity of their event types gives the impression that it might have been possible to do another round of aligning between the two annotators on the exact definitions of events to be annotated. A bit more discussion of why it wasn’t feasible to get higher agreement might help build confidence in the usefulness of that part of the dataset.
>
> We appreciate the reviewer's observation regarding the discernible disparity in inter-annotator agreement (IAA) between entity and event annotations. Nonetheless, considering that IAA is strictly a measure of whether the annotation indexes align exactly. it has the potential for enhancement when accounting for instances of partial agreement between annotators.
>
> On the other hands, annotations for event extraction (EE) task is more difficult than that for named entity recognition (NER) task for the following reasons:
>
> * **Complicated text structure** : It needs to consider inter-sentential relationship since relevant information of an event often spans multiple sentences.
> * **Interactions between event elements** : While NER often deals with tagging of individual entities, EE requires tagging based on semantic interpretations among event trigger and arguments such as Cause and Object.
>
> Therefore, we've struggled our best, but there's still some improvements. Before we release the dataset, we will make one more round of enhancements by post-processing.
>
> > The model experiments at the end of the paper are also not described in much detail. In this case, that is probably fine, since the authors are reproducing existing models from past research, and the focus of their paper is on the new dataset rather than any model innovations. They show results for two models each for the NER and EE tasks, respectively. There isn’t a lot to conclude from those findings, though, other than that the models do seem to be learning.
>
> In accordance with the reviewer's comment, we will add implementation details to the appendix in the camera-ready. We adopted their best hyper-parameters of the NER and EE models reported in the original paper. We trained the models with one NVIDIA RTX 3090 GPU.
>
> > I’m not entirely sure how to assess whether these results are impressive or not, because no comparative results from similar research are shown. The authors are the first to use their own dataset. But since their goal is to demonstrate the value of their new dataset, it actually might be helpful to show how the same types of models performed on a previous dataset. If the authors get at least similar scores when the models are trained on their own dataset, that could indicate that their dataset is as consistently labeled and useful for training NER and EE models as other datasets have been.
>
> The objective behind evaluating recent NER and EE models on our dataset is two-fold: firstly, to facilitate our dataset, thereby alleviating the burden on researchers and secondly, to address the potential interpretational challenges encountered by users who are introduced to our dataset for the first time. To train the models, we used only publicly verified open-source code and reproduced experiments on the datasets reported in previous studies to ensure that there are no implementation issues with the models. Then, we evaluated our dataset against the latest NER and EE models to provide baseline scores.

---

### Official Review · Reviewer_tSdi · 2023-07-31

**Typos Grammar Style And Presentation Improvements:** 1). Figure 1 should be large enough s…
**Soundness:** 3

**Excitement:**

3: Ambivalent: It has merits (e.g., it reports state-of-the-art results, the idea is nice), but there are key weaknesses (e.g., it describes incremental work), and it can significantly benefit from another round of revision. However, I won't object to accepting it if my co-reviewers champion it.

**Missing References:**

NA

**Paper Topic And Main Contributions:**

This work contributes a dataset for Event Extraction (EE) in the biomedical domain. EE is widely used to extract complex structures representing biological events from the literature. Complementing previous pieces of work, the proposed event extraction dataset concentrates on the animal experiment stage. The work establishes a novel animal experiment customised entity and event scheme in collaboration with domain experts to create an expert-annotated high-quality dataset containing discontinuous entities and nested events and evaluate recent outstanding named entity recognition and EE models on the dataset to establish benchmarks.

**Questions For The Authors:**

1). Lines 204-208: “...encompassing various entity types, especially at the cellular level. Given our primary focus on animal experiments, we consolidate these varied types into a single unified category named Anatomy.” This looks to me as if you are moving from fine-grained types to coarse-grained types. Have you looked into the impact of this on the performance of models? Don’t you think that it’s right for the models to learn fine-grained types, as these can sometimes actually affect the type of event being mentioned in text?

2). Lines 338 - 340: “We consider the annotation labels of the Ph.D. candidate as the ground truth and compare them to the other one’s.” This seems very biassed. Were there further assessments of one’s knowledge, experience, etc. taken into account in addition to just being a Ph.D. or Masters student?

3). Lines 381 - 383: “...we split the document into smaller chunks.” How did you decide on the size of the chunks? In a context where the context needed spans multiple sentences and these sentences fall into different chunks, how do you handle this? Eventually, how you perform chunking has a direct impact on the quality of the representations that the model learns, which affects the performance.

4). In your evaluation as described in lines 384–400, when identifying arguments, do you consider an argument correctly identified if both its span boundary and role are correct?



**Reasons To Accept:**

1). They contribute to the community an event extraction dataset concentrated on the animal experiment stage. They provide a detailed description of how the dataset was annotated. They further provide descriptions of the dataset and various statistics on its contents.

2). They run existing work in the field on the dataset to establish benchmarks.


**Reasons To Reject:**

See the questions below.

**Reproducibility:**

3: Could reproduce the results with some difficulty. The settings of parameters are underspecified or subjectively determined; the training/evaluation data are not widely available.

**Reviewer Confidence:**

5: Positive that my evaluation is correct. I read the paper very carefully and I am very familiar with related work.

---

> ### Author Rebuttal · Authors · 2023-08-29
>
> We sincerely appreciate the thoroughness invested in your review. We extend our gratitude for all constructive comments. To enhance readability, we have noted the comments in block quotes along with the corresponding responses below:
>
> > 1). Lines 204-208: “...encompassing various entity types, especially at the cellular level. Given our primary focus on animal experiments, we consolidate these varied types into a single unified category named Anatomy.” This looks to me as if you are moving from fine-grained types to coarse-grained types. Have you looked into the impact of this on the performance of models? Don’t you think that it’s right for the models to learn fine-grained types, as these can sometimes actually affect the type of event being mentioned in text?
>
> We appreciate the discerning observation made by the reviewer regarding the transition from fine-grained to coarse-grained entity types. Indeed, **this transition has been made to align with our domain focus on animal experiments**. This schema refinement was strategically designed in collaboration with experts. In discussions with experts, it was determined that the unified “Anatomy” category would be sufficiently helpful for animal experiments, rather than the cellular experiments that have been the focus of previous studies.
>
> > 2). Lines 338 - 340: “We consider the annotation labels of the Ph.D. candidate as the ground truth and compare them to the other one’s.” This seems very biassed. Were there further assessments of one’s knowledge, experience, etc. taken into account in addition to just being a Ph.D. or Masters student?
>
> We acknowledge the misleading statement. We reported precision, recall, and f1 score to **identify annotators' agreement between Ph.D. and Masters student**, rather than assigning the Ph.D students' annotations as the ground truth solely based on degree. We will revise the statement in the camera-ready.
>
> > 3). Lines 381 - 383: “...we split the document into smaller chunks.” How did you decide on the size of the chunks? In a context where the context needed spans multiple sentences and these sentences fall into different chunks, how do you handle this? Eventually, how you perform chunking has a direct impact on the quality of the representations that the model learns, which affects the performance.
>
> We acknowledge the potential ambiguity of our wording, which is split into smaller chunks rather than **truncation**. To provide a further clarity, the documents were truncated when they exceeded the predefined maximum input length of the model. In line with the reviewer’s comment,  we endeavored to minimize truncation for sufficiently contextualized inputs to facilitate event extraction. Notably, special consideration was granted to avoid fragmenting event trigger and their corresponding arguments of each event. We will add a detailed truncation process in the camera-ready.
>
> > 4). In your evaluation as described in lines 384–400, when identifying arguments, do you consider an argument correctly identified if both its span boundary and role are correct?
>
> Yes, the argument was considered correctly identified if both its text span boundary and role class were predicted correctly.
>
> We will revise all the typos and presentation improvements in the camera-ready.

---

### Official Review · Reviewer_zNXM · 2023-08-03

**Soundness:** 2

**Excitement:**

3: Ambivalent: It has merits (e.g., it reports state-of-the-art results, the idea is nice), but there are key weaknesses (e.g., it describes incremental work), and it can significantly benefit from another round of revision. However, I won't object to accepting it if my co-reviewers champion it.

**Paper Topic And Main Contributions:**

This paper describes a new corpus with annotated entities and events related to animal experiments. The work defined a specific set of entity types and event types according to the requirements in this topic. The paper does not give details of the annotation guidelines and focuses on dataset statistics and on results of applying selected NER and event extraction systems on this corpus.

**Reasons To Accept:**

The work describes a new annotated corpus on the specific topic of animal experiments.

**Reasons To Reject:**

The paper lacks details on the annotation process and guidelines.
The authors use NER and EE systems to "evaluate" the dataset, but this analysis does not actually show the quality or validity of the dataset.
The section on related work should focus more on event extraction datasets and tasks instead of including other works that do not directly align with the dataset and tasks described such as SemEval 2017 Task 10 and the ShARe13 and CADEC datasets.

**Reproducibility:**

2: Would be hard pressed to reproduce the results. The contribution depends on data that are simply not available outside the author's institution or consortium; not enough details are provided.

**Reviewer Confidence:**

3: Pretty sure, but there's a chance I missed something. Although I have a good feel for this area in general, I did not carefully check the paper's details, e.g., the math, experimental design, or novelty.

---

> ### Author Rebuttal · Authors · 2023-08-29
>
> We sincerely appreciate the thoroughness invested in your review. We extend our gratitude for all constructive comments. To enhance readability, we have noted the comments in block quotes along with the corresponding responses below:
>
> > The paper lacks details on the annotation process and guidelines.
>
> The annotation process is described in **Section 3.3. Annotation Procedure and Strategy**, and the annotation guideline has been made accessible through the provided link.
> https://docs.google.com/document/d/1BBl4mP9fJzYIRt5zOkEMLhM7__PvfjmDem33Tc1HHoU/edit?usp=sharing
>
> > The authors use NER and EE systems to "evaluate" the dataset, but this analysis does not actually show the quality or validity of the dataset.
>
> Inter-annotator agreement (IAA) has conventionally been the standard to indicate the dataset quality ([1], [2]). We also measured the quality of our dataset with IAA, which is described in **Section 3.4 Dataset Statistics and Characteristics-IAA** of the paper.
>
> [1] Gavin et al., 2023, Consistency is Key: Disentangling Label Variation in Natural Language Processing with Intra-Annotator Agreement
>
> [2] Jean Carletta, 1996, Assessing agreement on classification tasks: the kappa statistic.
>
> > The section on related work should focus more on event extraction datasets and tasks instead of including other works that do not directly align with the dataset and tasks described such as SemEval 2017 Task 10 and the ShARe13 and CADEC datasets.
>
> We acknowledge the observation that the content of our related work section might not be directly aligned with the event extraction datasets. We appreciate your feedback in this regard. In light of this, we intend to refine the related work section.

---

### Meta-Review · Area_Chair_sLAt · 2023-09-19

**Recommendation:** 3

**Metareview:**

This data paper describes a new gold dataset for biomedical NER and EE focused on scientific literature about animal experiments. The dataset  fills a gap in biomedical NLP.  The corpus is annotated for events and entities, according to a schema specifically designed consulting domain experts. The paper offers a description of the corpus creation process, proposes a new annotation schema,  and reports rich descriptive statistics. The two-stage annotation process followed can be robust enough to guarantee quality, provided the annotation guidelines are made clear. IAA is reported for both entities and events. Various existing NER and EE models are also evaluated with the new dataset, and the results support the efficacy of the dataset for NER and EE.
Both reviewers and authors engaged seriously in reviewing and discussion periods. The authors' responses were accurate and detailed; they acknowledged some of the indicated shortcomings, and provided clarifications and details on many of the points raised. Reviewers generally appreciated the work, but found some serious weaknesses in the original paper.  Particularly in need for attention are the following:
- an explanation of the differences in inter-annotator  agreement between entities and events is needed. Given the scores, a discussion seem to be beneficial as to whether and how far disagreement is intrinsic to the task or may depend on the annotation guidelines;
- having just two annotators questions the overall validity. This choice should be well-motivated;
- the complete annotation guidelines were made available only during the discussion phase, and still seem weak in some parts;
- the paper is well-written, but the framing of is not fully convincing, for instance “related works” could be more focused on the specific task addressed in the paper;
- the work would benefit from a comparison with performance on different datasets in the experimental part.

---

### Decision · Program_Chairs · 2023-10-07

**Decision:**

Accept-Findings

**Comment:**

This data paper describes a new gold dataset for biomedical NER and EE focused on scientific literature about animal experiments. The dataset  fills a gap in biomedical NLP.  The corpus is annotated for events and entities, according to a schema specifically designed consulting domain experts. The paper offers a description of the corpus creation process, proposes a new annotation schema,  and reports rich descriptive statistics. The two-stage annotation process followed can be robust enough to guarantee quality, provided the annotation guidelines are made clear. IAA is reported for both entities and events. Various existing NER and EE models are also evaluated with the new dataset, and the results support the efficacy of the dataset for NER and EE.
Both reviewers and authors engaged seriously in reviewing and discussion periods. The authors' responses were accurate and detailed; they acknowledged some of the indicated shortcomings, and provided clarifications and details on many of the points raised. Reviewers generally appreciated the work, but found some serious weaknesses in the original paper.  Particularly in need for attention are the following:
- an explanation of the differences in inter-annotator  agreement between entities and events is needed. Given the scores, a discussion seem to be beneficial as to whether and how far disagreement is intrinsic to the task or may depend on the annotation guidelines;
- having just two annotators questions the overall validity. This choice should be well-motivated;
- the complete annotation guidelines were made available only during the discussion phase, and still seem weak in some parts;
- the paper is well-written, but the framing of is not fully convincing, for instance “related works” could be more focused on the specific task addressed in the paper;
- the work would benefit from a comparison with performance on different datasets in the experimental part.